# Citizen Science in Biomedicine: Attitudes, Motivation, and Concerns of the General Public and Scientists in Latvia

**Alise Svandere** [1]**, Signe Mežinska** [2] **, Jekaterina Kaleja** [1]**, Normunds Kante** [1]**, Raitis Peculis** [1]**, Olesja Rogoza** [1] **and Vita Rovite** [1,*]

[1] Latvian Biomedical Research and Study Centre, Ratsupites Str. 1-k1, LV-1067 Riga, Latvia; alise.svandere@gmail.com (A.S.); katrina.kaleja@gmail.com (J.K.); normunds.kante@biomed.lu.lv (N.K.); raitis@biomed.lu.lv (R.P.); olesja.rogoza@biomed.lu.lv (O.R.)

[2] Institute of Clinical and Preventive Medicine, University of Latvia, Rainis Boulevard 19, LV-1586 Riga, Latvia; signe.mezinska@lu.lv

[*] Correspondence: vita.rovite@biomed.lu.lv

**Abstract:** Citizen science is research carried out by citizens in cooperation with scientists based on scientifically developed methods. Citizen science makes science accessible to the public and promotes public trust. Since there is scarce evidence about attitudes toward citizen science in the field of biomedicine, we aimed to evaluate the attitudes, motivations, and concerns of the Latvian general population and scientists from the biomedical research field toward citizen science research projects. We developed a survey that consisted of seven different citizen science research project examples (vignettes) and circulated it among the Latvian general population and researchers online, collecting quantitative and qualitative data. In total 314 individuals from the general population and 49 researchers filled in the survey. After the analysis was performed, we concluded that the general population and biomedical scientists in Latvia have different expectations toward citizen science. The results showed that while the general public is more interested in individual and societal benefits and concerned with specific participation aspects like filming, photographing, or co-funding, the scientists see the biggest potential contribution to their project in aspects of additional data collection and potential financial support, and are concerned about data quality, potential legal issues, and additional coordination communication that would be needed.

**Keywords:** citizen science; participation; knowledge production; biomedical research; research ethics; attitudes; motivation; concerns

## 1. Introduction

UNESCO's "Recommendation on Open Science" emphasizes the importance of opening the processes of scientific research to societal actors beyond the traditional scientific community (UNESCO Recommendation on Open Science 2021). One of the approaches embodying this vision is citizen science, defined by UNESCO as "models of scientific research conducted by non-professional scientists, following scientifically valid methodologies and frequently carried out in association with formal, scientific programmes or with professional scientists with web-based platforms and social media, as well as opensource hardware and software (especially low-cost sensors and mobile apps) as important agents of interaction" (UNESCO Recommendation on Open Science 2021). Broadly defined, citizen science includes the participation of laypeople at any phase of a research project (Fiske et al. 2019). These rather broad definitions open up opportunities for various forms of citizen science practices in biomedicine where the examples of citizen science projects are, e.g., gamification of data analysis as in Stall Catcher project (Nugent 2021), collecting and donation of biological samples and crowdsourcing as in the American Gut project (McDonald et al. 2018), co-designing of the study, participation in data analysis, co-drafting of research papers and policy recommendations as in Step Change project (Shah et al. 2023),

using of mobile apps for data collection and generating large biomedical datasets (Schmitz et al. 2018), and many other applications.

Citizen science can produce numerous benefits for society, the scientific community, and even individual citizen scientists which are broadly discussed in scientific literature, like the development of citizen scientists' knowledge and skills, increasing science literacy (Aristeidou and Herodotou 2020), maintaining physical fitness (Jurak et al. 2022), etc. At the same time, practicing citizen science may also lead to certain risks, e.g., problems in the quality of data, the safety of personal data, risks to animals, plants, ecosystems, or cultural heritage, the exploitation of citizen scientists (Rasmussen and Cooper 2019). To plan and implement citizen science projects, crucial prerequisites are the trust and willingness of the general public to participate, as well as the acknowledgment of the ethical aspects of citizen science (Resnik et al. 2015). To implement empirically informed ethical reflection and ethics oversite of citizen science projects, it is also important to analyze the attitudes, motivations, and concerns of the main stakeholders—both potential or practicing citizen scientists and researchers.

Despite the numerous quantitative and qualitative research studies exploring public attitudes toward citizen science in fields of biology, environmental science, digital humanities, and others (Martin et al. 2016; Bruckermann et al. 2021; Cigarini et al. 2022; Galanos and Vogiatzakis 2022; Greving et al. 2023), few studies have explored attitudes toward citizen science in the field of biomedicine. At the same time, citizen science in biomedicine has specific characteristics and challenges that might influence attitudes, motivations, and concerns of potential citizen scientists, e.g., collection of biological samples, processing of health, biometric and genetic data. The existing studies vary in the thematic, scientific field, and scale from research looking at the attitudes of participants of specific citizen science projects in biomedicine, e.g., a survey exploring the motivation of the British Gut project participants (Del Savio et al. 2017) to national level studies, e.g., a study analyzing the potential of citizen science in Switzerland, specifically looking also to field of medicine (Füchslin et al. 2019). There are also some more general studies, e.g., a survey of the general public on attitudes toward citizen science in life sciences has been conducted in the Czech Republic, Germany, Italy, Spain, Sweden, and the UK (Lakomý et al. 2020). Nevertheless, the data are fragmented, and specifically to our knowledge there are no studies exploring attitudes toward citizen science in Eastern Europe. The lack of data on Eastern Europe was one of the main reasons for conducting a study on this topic in Latvia.

The aim of our study was to evaluate the attitudes, motivations, and concerns of the general population and scientists from the biomedical research field in Latvia toward citizen science research projects in biomedicine. Our study showed that the general population and biomedical scientists in Latvia have different expectations from the citizen science. While the general public is more interested in individual and societal benefits and concerned with specific participation aspects like filming, photographing, or co-funding, the scientists see the biggest potential contribution to their project in aspects of additional data collection and potential financial support and are concerned about data quality, potential legal issues, and additional coordination communication that would be needed.

## 2. Methods

### 2.1. Questionnaire Development

The questionnaire was developed by the authors of the manuscript after conducting a literature analysis on citizen science projects already implemented in the field of biomedicine. The questionnaire consisted of seven vignettes including descriptions of potential citizen science projects and questions related to them and citizen science in general. Both target groups of this study, the general public and biomedical researchers, received the same vignettes. However, the questions related to the vignettes were different for each group of respondents. The questionnaire for the general public mainly included questions about attitudes and motivations to participate in each type of citizen science project, as well as about concerns related to participation. Scientists were asked about their attitudes,

motivations to conduct each type of citizen science study, and their concerns regarding the engagement of citizen scientists.

The survey was implemented via an online LimeSurvey form. The questionnaire for the general public was available in both Latvian and Russian, but the survey for researchers was available only in Latvian. For the general public, we included the option to complete a survey in Russian since approx. 30% of the Latvian population is Russian-speaking and part of this group, especially the elderly, have lower Latvian language skills. We did not translate the survey of researchers to Russian, as all academic work in Latvia is performed in Latvian and all scientists are able to complete the survey in Latvian. The questionnaire was tested by independent volunteers from both target groups, laypeople and researchers, representing both language communities, Latvian and Russian. The suggestions from piloting the questionnaire were implemented in the final version of the questionnaire.

### 2.2. Content of the Questionnaire

At the beginning of the questionnaire, the participants were given information about the aim of the survey, general information about citizen science, and examples of existing citizen science projects, e.g., bird watching or nature data collection. Additional information was given about the project in the framework of which this survey is implemented and about voluntary participation, rights of research participants, anonymity, and processing and protection of the collected data.

The questionnaire included vignettes describing seven different examples of potential citizen science projects in biomedicine. The full content of the surveys is included in Additional Files S1 and S2. Vignettes were designed to demonstrate various levels and types of participant involvement as that might influence people's motivations and concerns regarding participation. For example, vignettes include different types of data collection, sample donation, taking photos or videos, and the presence or lack of personal benefit to the participant. Two vignettes specifically included gamification and two included co-funding. The content of the vignettes can be summarized as follows:

1. *Research on Alzheimer's disease.* The citizen science activity is developed as a digital game with the aim of researching Alzheimer's disease. To participate, a citizen scientist should download the game on his/her mobile device and play the game to analyze imaging data derived from mice models to identify clogged blood vessels.
2. *Gut microbiome project.* The aim of the project is to study the microbiome in the general population. To participate, a citizen scientist should donate blood, collect and donate a fecal sample, and complete a dietary questionnaire. Afterward, each participant would receive personalized results about the composition of their gut microbiome. Participation includes co-funding of sample testing.
3. *My heritage.* The project offers to perform DNA analysis and obtain results about the participant's family history, ethnic background, and risks of developing certain diseases. The participant needs to collect their saliva sample and mail it to the laboratory. Participation includes co-funding of sample testing.
4. *Long-term consequences of COVID-19.* The project aims to investigate the long-term consequences of COVID-19. Participants should download an application on their mobile device, create their profile, and enter information about their health for a period of time.
5. *Movement analysis.* The aim of the study is to evaluate the health status of a person in a home setting. Participants should take a video of themselves doing several physical exercises, upload the video to the project website, and complete questionnaires. Participants would receive the results of their movement analysis and health recommendations.
6. *Wound healing.* The aim of the study is to investigate skin wound healing depending on treatment. The participant should take photos of the wound and upload them to a study website. Additionally, the participant needs to complete a questionnaire about wound treatment and healing.

7.   *Happiness project*. The project aims to study the psychological and neurological aspects of happiness. The participants should download an application on their mobile device, create their profile, and play four 20 min games, then repeat playing the games after several weeks.

In addition to questions, following the vignettes (Additional Files S1 and S2), socio-demographic data and other information about participants were collected via additional questions about age, gender, ethnicity, education level, and their previous experience with citizen science projects.

### 2.3. Data Collection

The survey was developed using the free online tool LimeSurvey Community Edition (version 5.4.7+221019). The survey was placed on the server at the Latvian Biomedical Research and Study Centre using the operating system Ubuntu version 22.4. Participants could complete the survey using an online webpage on their mobile device or computer through a specific survey link. Data were stored in MySql database and for the data analysis it was exported and converted to MS Excel for further processing. Data collection was done from 1 February until 1 April 2023. For the general public, an invitation to participate in the survey was disseminated via media and social networks. The link to the survey with an invitation to participate was published on the Latvian Biomedical Research and Study Center's website and Facebook page. The information was spread also on the radio and via other communication channels aiming to reach the broadest audience possible. For researchers, invitations to participate in the survey were sent via email lists of biomedical researchers and academic institutions in Latvia such as Latvian Biobanking Network, Latvian Young Scientist Association, and others. All collected data were completely anonymous and with no possibility to identify participants.

### 2.4. Data Analysis

After the closure of the surveys on 1 April 2023, the data file with responses was downloaded from the MySql database. All responses by the participants who had not completed the full survey were removed from the data collected and further data analysis was done only using the data from completely filled questionnaires. Initial data used for the analysis are available at https://osf.io/nm4rc/ (last access 1 April 2023). Quantitative data analysis was done in MS Excel. Descriptive statistics was performed for all study variables. Variables, numbers, and percentages were presented for categorical variables. In the questions where researchers needed to rank answers according to importance, rank points were awarded based on respondents' most important answers having a higher point amount. An inverse point scale was used ranging from 0 to N minus 1 where N is the number of answer choices per specific question.

For the analysis of qualitative data collected via open questions, we applied thematic analysis in three stages (Flick 2014). First, open coding was performed; second, all codes were reviewed and thematically similar codes were grouped into categories; third, the categories and codes were reviewed to ensure that there was no overlap between them. The number of open answers was limited (open questions were not obligatory and not all respondents provided them) and their lengths varied. In the coding process of answers by respondents representing the general public we identified two main categories—motivations and concerns. Table 1 shows categories and codes emerging from open answers of the general public. The codes above the line are the ones that emerged in all or multiple vignettes. The codes under the line were vignette-specific.

**Table 1.** Categories and codes.

| General Public | |
|---|---|
| **Motivations** | **Concerns** |
| • Opportunity to learn about the topic, for example, a disease<br>• Opportunity to learn about yourself<br>• Opportunity to be useful, to help scientists and society, and to develop the medical field<br>• Personal experience with the issue/topic<br>• Relevance and topicality of the issue<br>• Opportunity to follow one's health<br>• Entertaining participation<br>• Easy participation | • Lack of time or other resources<br>• Complicated participation<br>• Lack of knowledge, skills<br>• Need to make a co-payment<br>• Personal data security (data usage and storage)<br>• No understanding of benefits/purpose<br>• Unpleasant data collection (filming, taking photos of oneself, faecal collection)<br>• Dislike of computer games |
| • Fear of Alzheimer's disease | • Previous experience with DNA test<br>• Scepticism toward COVID-19<br>• Not being infected by COVID-19<br>• Unwillingness to do physical exercise<br>• Having no skin wounds<br>• Concerns about experimenting with treating of wounds |

## 3. Results

### 3.1. Socio-Demographic Characteristics of the Respondents

The survey for the general public was opened by 524 people and filled in completely by 314 respondents (Table 2). Most of the 314 participants had attained a higher level of education (75%) and most were Latvians (93%). Respondents represented all age groups, with slightly more participants in the age of 30–49. The gender division of the participants was unequal, with more women (86%) than men participating.

**Table 2.** Socio-demographic characteristics of respondents from general public.

| Variable | Category | General Public |
|---|---|---|
| Gender (N, %) | Male | 44 (14.0) |
| | Female | 270 (86.0) |
| Age (N, %) | 17–29 | 38 (12.1) |
| | 30–39 | 69 (22.0) |
| | 40–49 | 95 (30.3) |
| | 50–59 | 60 (19.1) |
| | 60+ | 52 (16.6) |
| Ethnicity (N, %) | Latvian | 292 (93.0) |
| | Russian | 15 (4.8) |
| | Other | 7 (2.2) |
| Education level attained (N, %) | Higher education | 235 (74.8) |
| | Occupational highschool | 43 (13.7) |
| | Highschool | 33 (10.5) |
| | Secondary school | 2 (0.6) |

The survey for scientists was opened by 107, but thoroughly filled in by 49 scientists (Table 3). Among them, most identified themselves primarily as researchers (86%), and less as academic staff (8%). The main fields of research represented were biology, medicine, pharmacology/chemistry; 10% of participants chose the answer "other". The target audience of the survey was researchers working in the field of biomedicine and the survey dissemination channels specifically targeted this group. Almost all participants of scientific background (94%) were in the age group 19–49, and 71% were females.

**Table 3.** Socio-demographic characteristics of scientists.

| Variable | Category | Scientists |
| --- | --- | --- |
| Gender (N, %) | Male | 15 (30.6) |
| | Female | 34 (69.4) |
| Age (N, %) * | 19–29 | 14 (29.8) |
| | 30–39 | 18 (38.3) |
| | 40–49 | 12 (25.5) |
| | 50–59 | 1 (2.1) |
| | 60+ | 2 (4.3) |
| Ethnicity (N, %) | Latvian | 40 (81.6) |
| | Russian | 6 (12.2) |
| | Other | 3 (6.1) |
| Education level attained (N, %) | Bachelor's degree | 5 (10.2) |
| | Master's degree | 11 (22.4) |
| | Doctor of philosophy | 28 (57.1) |
| | Occupational studies degree | 3 (6.1) |
| | Other level | 2 (4.1) |
| Field of education (N, %) | Biology | 33 (67.3) |
| | Medicine | 7 (14.3) |
| | Pharmacology/chemistry | 4 (8.2) |
| | Other | 5 (10.2) |
| Main occupation (N, %) | Scientist (leading researcher, researcher, research assistant, lab assistant) | 41 (83.7) |
| | Academic staff (docent, professor, lecturer, etc.) | 5 (10.2) |
| | Medical staff (doctor, medical support staff) | 2 (4.1) |
| | Other | 1 (2.0) |

* two researchers did not indicate their age.

### 3.2. Previous Experience with and Openness to Citizen Science

In this survey, we measured whether both groups, the general public and the scientists, had previous experience with citizen science projects (Table 4). Most of the participants of the general public (70%) (N = 219) answered that they had no experience with citizen science, but 18.5% (N = 58) had previous experience of participating in a citizen science project. The scientists were asked if they had been previously engaged with citizen science—either as participants or researchers. A total of 53% (N = 26) answered that they have not been engaged with citizen science before, but 41% (N = 20) of researchers reported having had an experience with citizen science. In addition, the scientists were asked if they had ever involved citizen scientists in their research projects. A total of 82% (N = 40) reported that they had not engaged citizen scientists in their research studies, however, 63% (N = 31) of all researchers who participated in the survey would consider this idea.

**Table 4.** Previous experience with and openness to citizen science.

| | Question | Answer | N, % |
| --- | --- | --- | --- |
| General public | Have you ever participated in a citizen science project (for example, collecting nature data, microbiome research, deciphering folklore records)? | Yes | 58 (18.5) |
| | | No | 219 (69.7) |
| | | Don't know/don't remember/not sure | 37 (11.8) |
| Scientists | Have you ever participated in a citizen science project (for example, collecting nature data, microbiome research, deciphering folklore records)? | Yes | 20 (40.8) |
| | | No | 26 (53.1) |
| | | First time hearing about citizen science | 2 (4.1) |
| | | I am planning to take part | 1 (2.0) |
| | Have you ever engaged citizen scientists in your research? | Yes | 8 (16.3) |
| | | No | 40 (81.6) |
| | | I don't know | 1 (2.0) |
| | Would you like to engage citizen scientists in your research? | Yes | 31 (63.3) |
| | | No | 4 (8.2) |
| | | I don't know | 14 (28.6) |

*3.3. Motivation*

After each vignette, we asked the general public about their motivation to participate in a study like this, and the scientists were asked about their interest in using a similar approach in their scientific work.

The two projects described in vignettes in which participants from the general public were most motivated to participate (see answer *Definitely*) were the DNA study *My heritage* (40.8%) and the *Gut microbiome* projects (29.0%). Both of these studies included co-payment and individual benefits to the participant (Table 5). The survey results show that the respondents see individual benefits as the main motivation to participate in these two projects. Open answers showed that many respondents are personally interested in topics of the *My heritage* and the *Gut microbiome* projects. The respondents also described what they imagined the potential benefits to be—learning about their family, genetic diseases, gut microbiome, etc.

The two least popular projects from the point of view of the general public (see answer *Definitely not*) were *Movement analysis*, including video recording of physical exercise (11.5%), and *Wound healing*, involving taking photos of skin wounds (7.3%). These vignettes did not include co-funding or crowdsourcing, but both required sharing of video or photo materials of oneself with researchers. Like with the two most popular vignettes, also for the *Movement analysis* project, 81.0% of respondents valued individual benefits of participation—receiving analysis of their physical exercise and health advice. At the same time, in open responses they reported to be concerned about the lack of time for participation and their personal data security. The open answers also showed that some people would feel highly exposed by filming themselves and knowing that somebody would later watch and analyze the video.

The researchers preferred the DNA study *My heritage* (63.3%), the *Long-term consequences of COVID-19* (63.3%), and the *Gut microbiome study* (59.2%) as approaches they might use in their scientific work. In the open answers researchers commented that they see an opportunity to collect useful data for their scientific work via projects like these. They perceived the suggested methods as an effective way to collect data and the type of engagement was viewed as convenient for the participants since it does not require specific knowledge in the field and in two cases also does not take a lot of time.

As less relevant to their research scientists considered *Movement analysis* including physical exercise video recording (55.1%), *Wound healing* involving taking photos of wounds (42.9%), and the *Happiness project* involving mental health data collection (42.9%) (Table 5). *Wound healing* and the *Happiness project* were described in open answers as not so much related to the scientists' field of research and having a less convincing research design. *Movement analysis* was also seen as including a less relevant approach to data collection. In addition to that, the means of participation was described as complex, and there were concerns regarding following instructions and the data quality.

Individual benefits were indicated as the most important motivation to participate by the general public in vignettes that included feedback on individual results (*Gut microbiome study*, *My heritage*, and *Movement analysis*) (Table 6). In vignettes without individual benefit, respondents chose the benefit for society as the most important motivation and contribution to science as the second one. The least important motivation to participate was "Interesting way to spend my free time", ranging from 6.6% to 38.7% for different vignettes (Table 6). However, it is significant that an interesting way of spending time was the most popular motivation for two vignettes that include gamification—the *Research on Alzheimer's disease* in which participants play a game to analyze imaging data, and the *Happiness project* in which the games are played for the purpose of studying psychological and neurological aspects of happiness.

**Table 5.** Readiness to participate and use for data collection.

| Description of Vignettes | General Public | | | | | Scientists | | |
|---|---|---|---|---|---|---|---|---|
| | How Likely Are You to Participate in Such a Project? (N, %) | | | | | Would it Be Relevant to Collect Data in This or Similar Way in Your Field of Science? (N, %) | | |
| | Defini-tely | Probably | Maybe | Probably Not | Defini-tely Not | Yes | No | I Don't Know |
| *Research on Alzheimer's disease* including gamification for data analysis in neuroscience *No individual benefit, no co-payment* | 90 (28.7) | 127 (40.4) | 64 (20.4) | 23 (7.3) | 10 (3.2) | 19 (38.8) | 19 (38.8) | 11 (22.4) |
| *Gut microbiome project* including collection of fecal samples *Individual benefit, co-payment* | 91 (29.0) | 80 (25.5) | 73 (23.3) | 56 (17.8) | 14 (4.5) | 29 (59.2) | 13 (26.5) | 7 (14.3) |
| *My heritage project* including DNA analysis and collection of saliva samples *Individual benefit, co-payment* | 128 (40.8) | 86 (27.4) | 53 (16.9) | 35 (11.1) | 12 (3.8) | 31 (63.3) | 14 (28.6) | 4 (8.2) |
| Survey-based project on *Long-term consequences of COVID-19 No individual benefit, no co-payment* | 76 (24.2) | 70 (22.3) | 63 (20.1) | 77 (24.5) | 28 (8.9) | 31 (63.3) | 12 (24.5) | 6 (12.2) |
| *Movement analysis* project aimed at development of diagnostic tool based on recording of physical exercise *Individual benefit, no co-payment* | 70 (22.3) | 66 (21.0) | 59 (18.8) | 83 (26.4) | 36 (11.5) | 18 (36.7) | 27 (55.1) | 4 (8.2) |
| *Wound healing project* on healing and care of wounds based on collection of survey and photographs *No individual benefit, no co-payment* | 41 (13.1) | 72 (22.9) | 81 (25.8) | 97 (30.9) | 23 (7.3) | 21 (42.9) | 21 (42.9) | 7 (14.3) |
| *Happiness project* including gamification for data collection on mental health *No individual benefit, no co-payment* | 104 (33.1) | 99 (31.5) | 53 (16.9) | 39 (12.4) | 19 (6.1) | 20 (40.8) | 21 (42.9) | 8 (16.3) |

**Table 6.** Motivation to participate of general public.

| Description of Vignettes | What Would Be Your Motivation to Participate in Such a Project? (N, %) | | | | | | | |
|---|---|---|---|---|---|---|---|---|
| | I See Benefits for Myself | | I See Benefits for the Society | | I Want to Contribute to Science | | Interesting Way to Spend My Free Time | |
| | Yes | No | Yes | No | Yes | No | Yes | No |
| *Research on Alzheimer's disease* including gamification for data analysis in neuroscience *No individual benefit, no co-payment* | 106 (37.7) | 175 (62.3) | 184 (65.5) | 97 (34.5) | 144 (51.2) | 137 (48.8) | 87 (31.0) | 194 (69.0) |
| *Gut microbiome project* including collection of fecal samples *Individual benefit, co-payment* | 191 (78.3) | 53 (21.7) | 126 (51.6) | 118 (48.4) | 119 (48.8) | 125 (51.2) | 16 (6.6) | 228 (93.4) |
| *My heritage project* including DNA analysis and collection of saliva samples *Individual benefit, co-payment* | 233 (87.3) | 34 (12.7) | 86 (32.2) | 181 (67.8) | 110 (41.2) | 157 (58.8) | 51 (19.1) | 216 (80.9) |
| Survey-based project on *Long-term consequences of COVID-19 No individual benefit, no co-payment* | 114 (54.5) | 95 (45.5) | 137 (65.6) | 72 (34.4) | 120 (57.4) | 89 (42.6) | 18 (8.6) | 191 (91.4) |
| *Movement analysis* aimed at development of diagnostic tool based on recording of physical exercise *Individual benefit, no co-payment* | 158 (81.0) | 37 (19.0) | 81 (41.5) | 114 (58.5) | 79 (40.5) | 116 (59.5) | 42 (21.5) | 153 (78.5) |
| *Wound healing project* on healing and care of wounds based on collection of survey and photographs *No individual benefit, no co-payment* | 95 (49.0) | 99 (51.0) | 109 (56.2) | 85 (43.8) | 118 (60.8) | 76 (39.2) | 17 (8.8) | 177 (91.2) |
| *Happiness project* including gamification for data collection on mental health *No individual benefit, no co-payment* | 161 (62.9) | 95 (37.1) | 148 (57.8) | 108 (42.2) | 135 (52.7) | 121 (47.3) | 99 (38.7) | 157 (61.3) |

The most significant motivation of the researchers (Table 7) for including elements of citizen science in their research studies was to collect a larger amount of data (99 rank points) with lower costs (55 rank points) and do this faster (44 rank points). The education of the society (43 rank points) and making science accessible (37 rank points) were considered less significant motivations.

**Table 7.** Motivation of scientists to engage citizen scientists in their research.

| Why Would You Like to Engage Citizen Scientists in Your Research? | Rank Points * |
|---|---|
| It is an opportunity to collect more data (attract more research participants) | 99 |
| It is an opportunity to collect data faster | 44 |
| It is an opportunity to collect data with less costs | 55 |
| To democratize science (make it more accessible to the general public) | 37 |
| To increase the public's level of understanding of my research field | 43 |

\* Higher rank points mean higher relevance to the scientists.

*3.4. Concerns*

The most important concern expressed by the general public regarding vignettes *Gut microbiome* project and *My heritage* where co-funding was required was the co-payment (43.6% and 44.6%) (Table 8). Asked how much participants would be willing to co-pay to participate in a particular study, for the *Gut microbiome* project, 55% of respondents indicated that they would be willing to pay up to 20 euros and 24% were ready to pay 21 to 50 euros; 17% indicated that they would not participate if there was a co-payment (Figure 1). For the *My heritage* project, 38% of respondents were ready to pay less than 20 euros, 34%—21 to 50 euros, and 14%—51 to 100 euros; 12% of respondents did not want to participate in this study if a co-payment is required. For these two vignettes (*Gut microbiome project* and *My heritage*), participants also less frequently chose the answer "I have no concerns regarding participating"—for the *Gut microbiome project*—26.1% and for the *My heritage*—37.9%. This was also the case of physical exercise recording vignette (35.4%). For all other vignettes this answer was positive for 49.0% up to 60.5%.

Biological sample donation raised concerns for 17.2% of participants in the case of fecal samples and 5.4% in the case of saliva. The biggest concerns for personal data security were indicated for the vignette *My Heritage* including DNA analysis (29%) and physical exercise recording (22.3%). Lack of time was indicated as a significant demotivator for most vignettes ranging from 18.5 to 29.3%, with the exception of the *My heritage* study, where lack of time was an issue only to 6.1% of respondents.

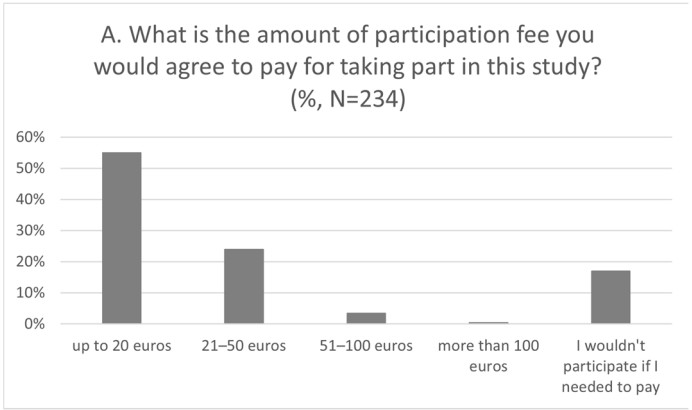 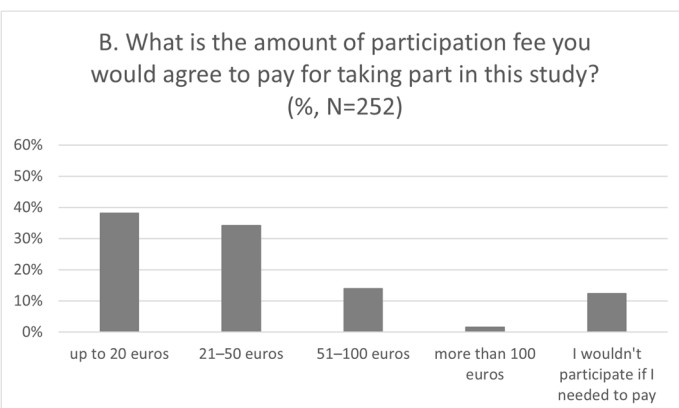

**Figure 1.** The amount of co-funding participants that were willing to pay for taking part in the (**A**) *Gut microbiome* study and (**B**) *My heritage* project.

**Table 8.** Concerns of the general public toward participation in citizen science projects.

| Description of Vignettes | What Would Be Your Concerns When Participating in Such a Project? (N, %) | | | | | | | | | | | | | | | | | |
|---|---|---|---|---|---|---|---|---|---|---|---|---|---|---|---|---|---|---|
| | My Personal Data Security | | Security Threats to My Electronical Device | | Lack of Time to Participate | | Necessity to Donate Biological Samples | | Concern for Research Results that Include Information about My Health | | Co-Payment | | Technical Issues with Video Recording and Uploading | | I Have No Concerns Regarding Participation | | Other | |
| | Yes | No | Yes | No | Yes | No | Yes | No | Yes | No | Yes | No | Yes | No | Yes | No | Yes | No |
| *Research on Alzheimer's disease* including gamification for data analysis in neuroscience *No individual benefit, no co-payment* | 66 (23.5) | 215 (76.5) | 54 (19.2) | 227 (80.8) | * | | | | | | | | | | 170 (60.5) | 111 (39.5) | 18 (6.4) | 263 (93.6) |
| *Gut microbiome project* including collection of fecal samples *Individual benefit, co-payment* | 54 (17.2) | 260 (82.8) | | | 66 (21.0) | 248 (79.0) | 54 (17.2) | 260 (82.8) | 49 (15.6) | 265 (84.4) | 137 (43.6) | 177 (56.4) | | | 82 (26.1) | 232 (73.9) | 15 (4.8) | 299 (95.2) |
| *My heritage project* including DNA analysis and collection of saliva samples *Individual benefit, co-payment* | 91 (29.0) | 223 (71.0) | | | 19 (6.1) | 295 (93.9) | 17 (5.4) | 297 (94.6) | | | 140 (44.6) | 174 (55.4) | | | 119 (37.9) | 195 (62.1) | 17 (5.4) | 297 (94.6) |
| Survey-based project on *Long-term consequences of COVID-19 No individual benefit, no co-payment* | 56 (17.8) | 258 (82.2) | 39 (12.4) | 275 (87.6) | 77 (24.5) | 237 (75.5) | | | | | | | | | 154 (49.0) | 160 (51.0) | 32 (10.2) | 282 (89.8) |
| *Movement analysis* aimed at development of diagnostic tool based on recording of physical exercise *Individual benefit, no co-payment* | 70 (22.3) | 244 (77.7) | 31 (9.9) | 283 (90.1) | 92 (29.3) | 222 (70.7) | | | | | | | 86 (27.4) | 228 (72.6) | 111 (35.4) | 203 (64.6) | 22 (7.0) | 292 (93.0) |
| *Wound healing project* on healing and care of wounds based on collection of survey and photographs *No individual benefit, no co-payment* | 48 (15.3) | 266 (84.7) | 22 (7.0) | 292 (93.0) | 81 (25.8) | 233 (74.2) | | | | | | | | | 159 (50.6) | 155 (49.4) | 45 (14.3) | 269 (85.7) |
| *Happiness project* including gamification for data collection on mental health *No individual benefit, no co-payment* | 66 (21.0) | 248 (79.0) | 50 (15.9) | 264 (84.1) | 58 (18.5) | 256 (81.5) | | | | | | | | | 173 (55.1) | 141 (44.9) | 17 (5.4) | 297 (94.6) |

\* In grey fields are questions that have not been asked after a specific vignette

The additional concern indicated in the open answers by the general public was negative attitude toward digital games. Although some of the respondents evaluated the *Research on Alzheimer's disease* and the *Happiness project* as an interesting way to spend their free time, for others, playing games was something they claimed to purposefully avoid in their daily activities. Gamification was perceived also as a less serious or trustful way of collecting data.

Scientists were most often concerned about a lack of control in data collection (253 rank points), followed by data protection and legal aspects (234 rank points) (Table 9). Higher concerns were also expressed toward difficulty publishing results (153 rank points), data quality and traceability (148 rank points), and the potential need for additional resources like communication or coordination (148 rank points). There were fewer concerns about ethical aspects (127 rank points) and the public's disinterest in participation (68 rank points).

**Table 9.** Concerns of scientists toward engaging citizen scientists in their research.

| What Would be Your Concerns When Engaging Citizen Scientists in Your Research? | Rank Points * |
|---|---|
| Quality and traceability of data | 148 |
| Lack of control in data collection process | 253 |
| Ethical aspects | 127 |
| Data protection and normative aspects | 234 |
| Public's disinterest in participation, low participation | 68 |
| Need for additional resources: coordination, communication with volunteers | 148 |
| Difficulty in publishing the obtained results in higher class journals | 153 |
| No concerns regarding engagement of citizen scientists in research | 98 |

* Higher rank points mean higher relevance to the scientists.

### 3.5. Scientists' Perception of the Role of Citizen Science

We asked the scientists to answer an open question about how they might use citizen science in their research and what is the role of citizen science in their field of science. The answers were generally in line with the quantitative data. Many scientists viewed citizen science as a tool to collect data more effectively (faster and in larger amounts) and to attract crowdfunding or co-payment for their research. Researchers also shared concerns about the quality and representability of the data collected by citizen scientists. The topicality of issues regarding the data collection and data quality reveals that the data, the quantity and quality, is a priority for scientists. Three other topics mentioned in answers to open questions were: adding novel societal perspectives, the promotion of science, and increasing trust in science via interactions between scientists and the general public. Scientists also mentioned the benefit for both biomedical research and society in general.

Some scientists emphasized that citizen science in biomedicine serves by improving both health and science literacy. It was also seen as an opportunity to inspire people to become more active as citizens. Nevertheless, it was pointed out that citizen science is quite an underdeveloped field in Latvia.

Most of the scientists who answered open questions on the role and potential use of citizen science saw it as potentially beneficial and of high importance. They acknowledged having little experience with this type of engagement of society in their research, but at the same time, most of them showed motivation to learn more about citizen science and to try using its methods in their scientific work.

### 4. Discussion

To our knowledge, this is the first study in Latvia and Eastern Europe on attitudes, motivations, and concerns of the general public and researchers toward citizen science in the field of biomedicine. Overall, our study provides an insight into views on citizen science in biomedicine from the two most important perspectives—the general public and researchers. The results might help to design new citizen science projects in the future, to

build trustworthiness and trust, as well as to develop communication campaigns for citizen science projects.

Our results show that the attitudes of the public and scientists in Latvia differ considerably, as biomedical scientists consider citizen science primarily as a way to collect more data and attract additional financial resources via co-funding or crowdsourcing, but the general public in our study was mainly motivated by individual benefits from the study and was hesitant if any financial contribution is required from them. The biggest concern of the general public among all suggested types of concerns was the co-payment required. Between 12 and 17% of the general public were not willing to participate in a study if there was any co-payment or crowdfunding involved. Those respondents who were open to contribute financially mostly indicated an amount from 20 to 50 euros as acceptable.

These results differ from some previous studies on the motivation of citizen scientists in biomedicine, e.g., the participants of the *British Gut project* reported two prosocial motivations as the most important: (1) "to participate in clinically useful research", and (2) "to help clinicians in the search of cures for medical conditions" (Del Savio et al. 2017). However, it should be noted that the attitudes of the participants of the *British Gut project* were explored after their participation, so only those members of the general public participated who were motivated and had made a decision to participate were involved. In our sample, only 18.5% of the general public participating in the survey had previous experience of participation in citizen science projects. A survey of the general public on attitudes toward citizen science in life sciences conducted in the Czech Republic, Germany, Italy, Spain, Sweden, and the UK showed that men and younger persons are more extrinsically motivated (e.g., by rewards or awards, improving self-image), but women and older people are more intrinsically motivated (e.g., being interested in the research topic, helping society, increasing the impact of research) (Lakomý et al. 2020). These differences between our and previous research studies show that attitudes, motivations, and concerns of the general public vary from population to population and are influenced by socio-cultural differences even in European countries. To build and maintain trust and trustworthiness in relationships between scientists and citizen scientists it is important to take into account local socio-cultural characteristics. For example, the fact that in the Eurobarometer 2021 survey, respondents from Latvia specifically mentioned the feeling they would not be welcomed or that it is "not something for them" as a barrier to engagement with science and technology should be taken into account when planning and implementing citizen science projects in Latvia (Directorate-General for Communication European Commission 2022).

The results of our study also provide important information for organizing communication campaigns for citizen science projects in biomedicine. On the one hand, it is clear that emphasizing individual benefits available in citizen science projects might help to attract more citizen scientists, at least in a short time. On the other hand, the survey results show that in Latvia, it is necessary to empower science literacy in society and to educate citizens about the role of science and the public benefits of scientific research. It goes in line with the results of the 2021 Eurobarometer survey where respondents from Latvia demonstrated lower trust in science and less desired public involvement in decisions about science and technology than the EU average (Directorate-General for Communication European Commission 2022).

Regarding the donation of biological samples, we observed that members of the general public were more concerned about the donation of fecal samples compared to saliva derived DNA. The logistics of fecal samples in the vignette was described to be performed through a diagnostic laboratory, while saliva was suggested to be sent by regular mail shipping. Presumably, the respondents were more concerned by the unpleasantness of the collection of the fecal sample than other considerations. Our data from a previous study regarding the willingness to donate different types of biological samples to biobanks in Latvia also indicated that the general public in Latvia is more open to donating blood (45.4%) compared to fecal samples (40.4%) (Mezinska et al. 2020).

Our study has several limitations. As the study population was not designed to be a representative group for the general population or biomedical scientists in Latvia the results reflect the opinions of the sample that filled in the survey with specific deviations in gender and ethnicity representation. Additionally, we discovered that for the first vignette about Alzheimer's disease data analysis, we did not include "*Lack of time*" as an answer to concerns of the general population regarding the project. This was an error in survey digitalization, but we have represented results for other vignettes and consider that this does not significantly impact the results of the study.

Sociodemographic characteristics of the general population participants of our study show significant differences in gender representation—86% of the respondents were women. This is in line with previous studies that women overall are more interested in participating in voluntary-based studies than men (Lakomý et al. 2020). We also did not implement a specific recruitment strategy to ensure the representability of certain population groups. As a result, the Russian-speaking population was less represented (4.8%) than it is in the general population of Latvia (~30%) (Iedzīvotāju skaits pašvaldībās pēc nacionālā sastāva 2022). Although the survey was translated and made available in Russian, we consider that the main reason for the lack of representation of the Russian-speaking population was the information channels through which we distributed the invitation. A potential strategy we would suggest that could improve both gender and ethnicity sample representativeness in future work is using targeted recruitment.

## 5. Conclusions

The general population and scientists in Latvia have different expectations regarding citizen science. The general public is more interested in individual and societal benefits and concerned about specific participation aspects like co-funding or the necessity to take and share photos and videos. The scientists see most contributions to their project in aspects of additional data collection and potential financial support, and are concerned about data quality, potential legal issues, and additional coordination and communication that would be needed. These differences between the attitudes of the general public and scientists should be taken into account when developing future citizen science projects in Latvia. Moreover, motivation to participate in such projects could be improved by educating the general population on the role of science in society in general.

The results of our study indicate that it is important to assess the attitudes toward citizen science in specific populations, as, for example, compared to studies in some other countries the most important motivation of the general public for involvement in citizen science activities in Latvia is an individual benefit and societal benefit is secondary.

**Supplementary Materials:** The following supporting information can be downloaded at: https://www.mdpi.com/article/10.3390/socsci12110620/s1, Additional File S1: Content of survey for general public; Additional File S2: Content of survey for researchers.

**Author Contributions:** Conceptualization, A.S., S.M., J.K. and V.R.; methodology, A.S., S.M., J.K., N.K., R.P. and V.R.; software, N.K.; validation, S.M. and V.R.; formal analysis, A.S., S.M., N.K. and R.P.; investigation, A.S., S.M., V.R.; resources, S.M., N.K. and V.R.; data curation, R.P. and V.R.; writing—original draft preparation, A.S., S.M. and V.R.; writing—review and editing, A.S., S.M., J.K., O.R. and V.R; visualization, A.S. and V.R.; supervision, S.M. and V.R.; project administration, V.R.; funding acquisition, V.R. All authors have read and agreed to the published version of the manuscript.

**Funding:** This research was funded by the European Regional Development Fund within the project „DECIDE—Development of a dynamic informed consent system for biobank and citizen science data management, quality control and integration" (1.1.1.1/20/A/047).

**Institutional Review Board Statement:** Not applicable.

**Informed Consent Statement:** The anonymous survey was performed in accordance with relevant national guidelines and regulations. Informed consent was obtained from all participants by providing clear and concise information about the research study, data processing and research participants' rights in the introduction to the questionnaire. The introductory part of the questionnaire

also included a statement explaining that consent is provided by virtue of completing the survey. An anonymous sociological survey does not require approval by a research ethics committee in Latvia according to the Law on the Rights of the Patients, Sections 10 and 11 (these sections require review by the research ethics committees only for clinical trials and studies processing personal data of patients) and according to the Ethics Guidelines of the Latvian Association of Sociologists (http://sociologija.lv/etika-2/lsa-kodekss/).

**Data Availability Statement:** The datasets generated and/or analyzed during the current study are available in the OSF repository at link: https://osf.io/nm4rc/ (last accessed 1 April 2023). Additionally, the datasets used and/or analyzed during the current study are available from the corresponding author on reasonable request.

**Conflicts of Interest:** The authors declare no conflict of interest.

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
