# Peer review of "Citizen Science in Biomedicine: Attitudes, Motivation, and Concerns of the General Public and Scientists in Latvia"

_socsci, doi:10.3390/socsci12110620_

Round 1
Reviewer 1 Report
Comments and Suggestions for Authors
The paper presents a very interesting topic of the attitudes of both general public and researchers to applying citizen science in the biomedicine studies. The authors presented the used methodology and collected responses in a great detail. The results clearly show the barriers and limitations of using the citizen science. The authors also discussed the limitations of their study which shows that they are aware of them and can help them in planning further studies. In my opinion the paper is well written and does not require any improvements.
In my opinion there is no need to make any major or minor changes. I would only be interested in authors' opinion why the gender divide in the number of respondends was so scewed towards women. The other interesting issues is the not mentioned question of the differences among genders in their opinions on the pros and cons of citizen science.
Comments on the Quality of English Language
English is fine.
Author Response
Dear Reviewer,
The following are our responses to your comments.
- "The paper presents a very interesting topic of the attitudes of both general public and researchers to applying citizen science in the biomedicine studies. The authors presented the used methodology and collected responses in a great detail. The results clearly show the barriers and limitations of using the citizen science. The authors also discussed the limitations of their study which shows that they are aware of them and can help them in planning further studies. In my opinion the paper is well written and does not require any improvements."
Response: Thank you very much for appreciating the work we have done in researching people's attitudes towards citizen science in biomedical field. We are glad to read you find our paper interesting and well written.
- "In my opinion there is no need to make any major or minor changes. I would only be interested in authors' opinion why the gender divide in the number of respondends was so scewed towards women."
Response: As we see it, one way of explaining the gender imbalance in our respondent group is as we do it in the article by referencing Lakomý et al. (2020): previous studies show that "women overall are more interested to participate in voluntary-based studies than men." (line 478-479) In this research, we did not aim to recruit a representative sample of respondents due to the limited amount of resources we had.
- "The other interesting issues is the not mentioned question of the differences among genders in their opinions on the pros and cons of citizen science."
Response: Thank you for pointing this out. We did consider looking into the possible gender differences in attitudes, motivations and concerns, but what we found out is that (a) regarding qualitative data we do not have a sufficient number of open answers to argue for any such differences and (b) in regards to quantitative data, none of the differences was of statistical significance. In addition, due to the fact that only 14% of the respondents were males, there was not enough data to analyse differences in opinions between gender groups. Nevertheless, it is indeed a very interesting aspect that we could potentially look into in future studies.
Thank you very much for taking the time to carefully read our article and for helping us improve it.
Reviewer 2 Report
Comments and Suggestions for Authors
Thank you for this important article. I found it very interesting to evaluate the attitudes, motivation and concerns of the Latvian general population and scientists from biomedical research field towards citizen science research projects in Latvia.
A few questions / comments and suggestions:
Based on my review of the document, here are my observations and recommendations:
- In line 25-80 the introduction section, consider briefly mentioning the specific gap this study aims to address - the lack of existing research on attitudes to citizen science in biomedicine in Eastern Europe. This will further establish the rationale and significance.
- For the vignettes used in the surveys, it could be helpful to include a bit more information on how they were developed and the rationale for the different elements included.
- In line 369-421 the discussion of gender imbalance in respondents, note this reflects trends in other citizen science research and that targeted recruitment could help balance this in future studies.
- In line 422-439 when mentioning the limitations around sample representativeness, the authors could suggest potential strategies to improve representativeness in future work, such as targeted outreach.
- In line 369-421 the discussion effectively covers differences between this and other research. To further situate this study, the authors could comment on how the Latvian and Eastern European context may have influenced the findings.
- In line 441-451 the conclusion concisely summarizes the key points. Consider adding a statement about the study’s implications for future citizen science initiatives in the region.
Comments on the Quality of English LanguageMinor editing of English language required.
Author Response
Dear Reviewer,
The following are our responses to your comments.
- "Thank you for this important article. I found it very interesting to evaluate the attitudes, motivation and concerns of the Latvian general population and scientists from biomedical research field towards citizen science research projects in Latvia."
Response: Thank you very much for appreciating our efforts to research attitudes, motivation and concerns regarding citizen science in Latvia.
- "In line 25-80 the introduction section, consider briefly mentioning the specific gap this study aims to address - the lack of existing research on attitudes to citizen science in biomedicine in Eastern Europe. This will further establish the rationale and significance."
Response: Thank you for this comment. We agree that pointing out the lack of existing research on attitudes towards citizen science in biomedicine in Eastern Europe is an important argument for conducting the study. We mention it in the introduction already (lines 70-72). However, we agree that this can be communicated more clearly which is why have now added a sentence (lines 72-73) which indicates that this aspect was one of the reasons why we chose to research this subject in Latvia.
- "For the vignettes used in the surveys, it could be helpful to include a bit more information on how they were developed and the rationale for the different elements included."
Response: Thank you for helping us improve the description of the methods applied in the study. We have added now in the text (lines 86-88) that our questionnaire was developed after conducting literature analysis on already existing citizen science projects in biomedicine. Also, we have added a text (lines 125-126) that explains why we chose to include vignettes that demonstrate various levels and types of participant involvement.
- "In line 369-421 the discussion of gender imbalance in respondents, note this reflects trends in other citizen science research and that targeted recruitment could help balance this in future studies."
Response: We agree that targeted recruitment could help improve the gender imbalance in respondents. We added a sentence about it in lines 485-487.
- "In line 422-439 when mentioning the limitations around sample representativeness, the authors could suggest potential strategies to improve representativeness in future work, such as targeted outreach."
Response: We agree that the gender imbalance in the sample of respondents is an important limitation of the study and that it could be improved, e.g., by using survey outsourcing services. In our case, it was a limitation of resources that prevented us from implementation of more effective recruitment and sampling methods. However, we believe that despite the gender imbalance in the sample this study is unique in the Eastern European context and provides useful insights into the topic.
- "In line 369-421 the discussion effectively covers differences between this and other research. To further situate this study, the authors could comment on how the Latvian and Eastern European context may have influenced the findings."
Response: Thank you for the comment! Indeed the geographical context of Eastern Europe has the specific context of attitudes towards science as indicated by Eurobarometer surveys (see lines 454-458). We added additional information about attitudes towards science in the lines 444-448.
- "In line 441-451 the conclusion concisely summarizes the key points. Consider adding a statement about the study’s implications for future citizen science initiatives in the region."
Response: Thank you for the comment, we have updated our conclusion. Please see lines 507-510.
- "Minor editing of English language required."
Response: Thank you for the comment! We have performed language editing to correct presentation of grammar.
Thank you very much for taking the time to carefully read our article and for helping us improve it.